# Entrepreneurial Drivers for the Development of the Circular Business Model: The Role of Academic Spin-Off

**Stefano Poponi** [1],*[ID]**, Gabriella Arcese** [1][ID]**, Enrico Maria Mosconi** [2][ID] **and Michelangelo Arezzo di Trifiletti** [3]

1   Faculty of Economics, Niccolò Cusano University, Via Don Carlo Gnocchi, 3, I-00166 Rome, Italy; gabriella.arcese@unicusano.it
2   Department of Economics, Engineering, Society and Business Organization, Tuscia University, Via del Paradiso 47, I-01100 Viterbo, Italy; enrico.mosconi@unitus.it
3   Embassy of the United States of America, 00187 Rome, Italy; michaelarezzo@gmail.com
*   Correspondence: stefano.poponi@unicusano.it

**Abstract:** Circular Economy represents today a new economic paradigm based on the environment and on the recovery of material. The pursuit of this change can be implemented through different policies with a top-down or bottom-up approach. Following the latter approach Spin-Offs, typically defined as "Science Based" companies, represent an alternative tool to promote technology transfer. In other words, they represent a bridge between the research and the production system. This part of the study is part of a larger and more complex project whose objective is to verify whether the development of research Spin-Offs and in particular academics, operating in the environment sector, or more generally sustainable, facilitate the transition from the classic model of linear economics to the innovative model of circular economics. The aim of the paper is to investigate how spin off enterprises can be a driver for the development of a Circular Business Model and to facilitate the transition from the classical model of linear economy to the new model of Circular Economy. At the methodological level, a multiple compared analysis was made between a sample of firms located in Lazio Region- Italy, that operates in the area of green economy Smart Specialization Strategy (S3). The analysis shows a rapid succession of variables that lead to the identification of four scenarios, deriving from the interconnection of the outcome: "closed loop", "open loop" and the presence or absence of Circular Economy practices. The result confirms that the Academic Spin-Offs can be a driver of Circular economy, as long as that fall within the IV scenario, characterized by the interconnection of an open loop system that works in a circular approach. The "High valorization of waste" represents the discriminant in this scenario, which allows to activate a cascade system in a multi-stakeholder perspective.

**Keywords:** circular economy; academic spin-off; eco-innovation; business model; sustainable environment

---

## 1. Introduction

The concept of Circular Economy was associated by the Ellen MacArthur foundation with a generic term for defining an economy designed to be able to regenerate itself [1]. In a circular system the material flows are of two types: The biological ones, able to be reintegrated in the biosphere, and the technical ones, destined to be revalued without entering the biosphere. The Circular Economy is therefore an economic system which reuses materials in subsequent production cycles, reducing waste to a minimum to make the most of it [2].

The process of transition towards the Circular Economy is progressively changing the dynamics of innovation management and business strategies of companies. The emergence of this new paradigm, in the light of recent provisions at European level, or the pursuit of sustainable development goals, lead to the construction of a systematic approach among economic actors and to a progressive change of the models of development that focuses on the environment.

The need to move from a linear to a circular model leads to the basic concept of the loop mechanism [3–7], in an approach promoting the combined use of enabling factors, such as the reuse of waste materials, and waste, use in other production cycles of waste and scrap [8–10]. This change can be achieved through the adoption of business models that encourage the transition towards closed cycles, in a virtuous application of the principles of the CE.

In fact, many methods and tools have been upgraded to support the "design or architecture" and "value creation" [11] with the aim of achieving a more resource-efficient system [2], capable of incorporating circular principles into the Business Model. This definition of CE from Ellen MacArthur Foundation as: "an industrial system that is restorative or regenerative by intention and design. It replaces the "end-of-life" concept with restoration, shifts towards the use of renewable energy, eliminates the use of toxic chemicals, which impair reuse, and aims for the elimination of waste through the superior design of materials, products, systems, and, within this, business models" [1] Despite a shared and harmonized framework that is progressively consolidating compared to the implementation of Circular Models, studies are not yet present in the literature which define the role of Spin-Off companies in the transition to this new system.

Spin-Offs are innovative companies integrated in the renewal process of university systems, oriented towards the management of services of academic entrepreneurship for the marketing of research results through the exercise of business activity. Spin-Off companies are, without a doubt, an alternative tool for promoting the transfer of knowledge and transfer of technology to the commercial and productive sector. They constitute an important level of competition, capable of promoting the economic development of a country that is not only in creating jobs, generating wealth and adding value to academic research [12], by offering technological solutions that can contribute to a better understanding of sustainable growth of the systemin which they operate.

As they consolidate their role within the economic system (their survival rate is radically strengthening [13]. Spin-Off companies are contributing to increasing environmental and sustainability performance through their capacity of generating innovation by promoting forms of change in business models.

Despite this, the studies that recognize the contribution of the spin-offs in the transition to the CE are scarce.

Considering the evolution that the EC is bringing about global scenarios, this study has as its main goal to understand if and how spin off enterprises can be a driver for the development of a Circular Business Model.

More detailed explanation is given in Section 2 describes the research methodology employed. Next, Section 3 which presents the materials and methods used for developing the paper, formulates research framework and define the questions and defines its contributions to CE studies; Section 4 presents the results and provides the discussion; Section 5 shows the conclusions, including implications and limitations of this study, proposing future lines of research.

## 2. Materials and Methods

This work aims to contribute to the debate on the Circular Economy through the description and comparison of Spin-Off Companies belonging to the field of the Smart Specialization Strategy (S3) of the Lazio Region of Italy, in the Green Economy Area, following the research process represented in the Figure 1.

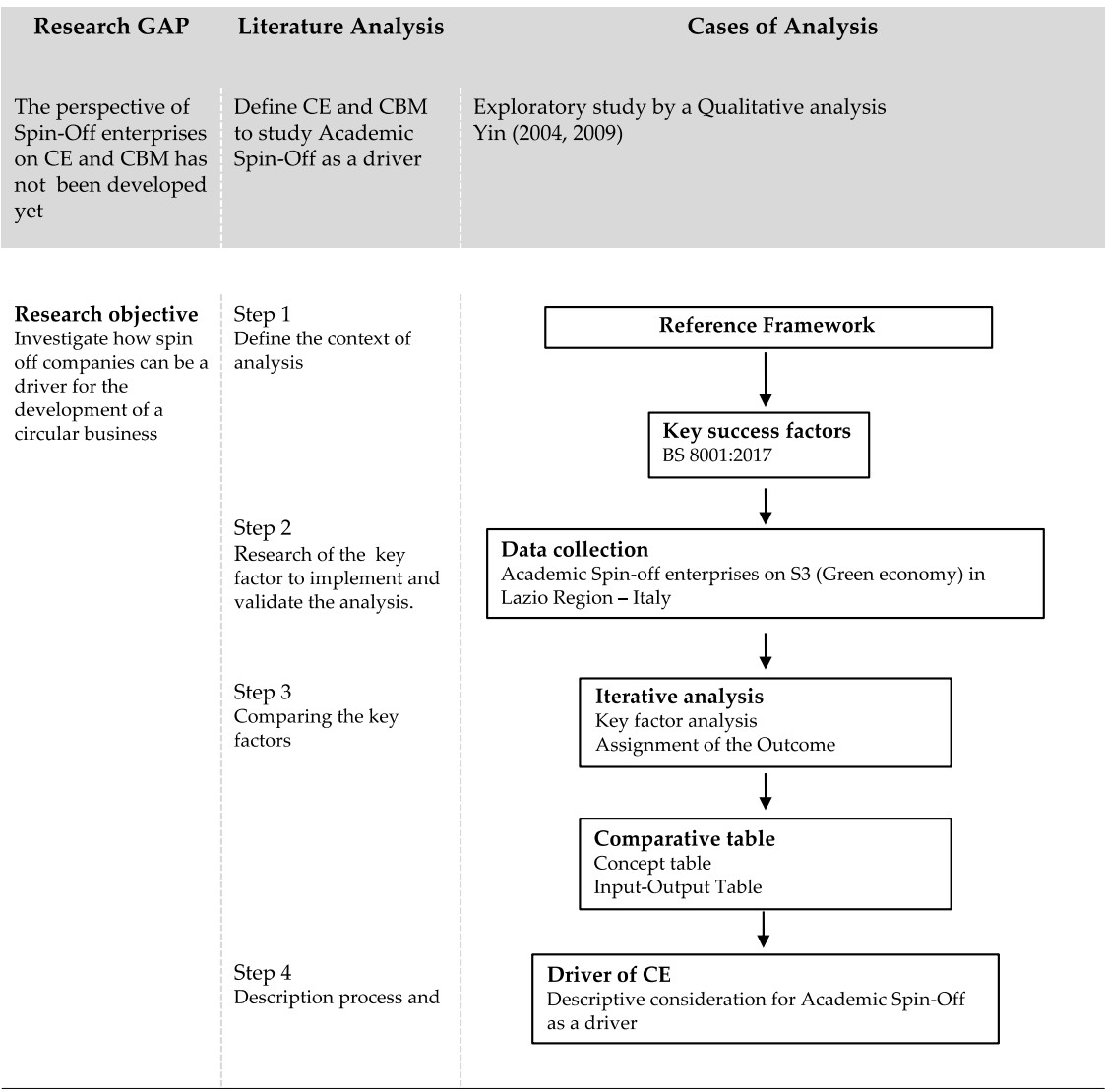

**Figure 1.** Research Framework. Source: Authors' Elaboration.

This research focus on the analysis of Academic Spin-Off and on their role in the transition of Circular Economy.

At a methodological level, several studies were analyzed for realized a reference framework, defined conceptual boundaries and investigated on the context and the state of art for this topic.

As shown in the Figure 1 to identify the research gap, a hybrid approach was used to analyze both the literature and the case studies composed by 4 steps.

In the first step the context was define though the factors that have an influence on the transition to the Circular Economy have been identified starting to the use of the new standard for Circular Economy: the BS 8001:2017. The standard provides a flexible management framework for implementing circular strategies inside the organizations. Starting from the principles of the CE, through 8 steps, it guides companies towards the implementation of a new Circular Business Model, based on "circularity maturity" [14]. The factors that the standard consider, and we use for the analysis are the following:

– Share. It considers the access or 'collaborative consumption' amongst users;
– Life extension. The lifetime of the product is function of the design of the products;
– Service support; it considers innovation support services, such as consultancy;
– Reduce, it relates to the environmental impact of the product or to the reduction of the number of components;

–　　Reuse/redistribute/repurpose. These consider the new destination of the product starting from a recycling management;

–　　Refurbish/remanufacture/recondition. These are focused to extend the life of the product, intervening on the aesthetic improvement, rebuilding or repairing to make it performant again.

With respect to the proposed factors, we add as a further element of analysis, that is the "High valorization of waste". It is "the non-obvious reuse involving high skills and/or technological innovation, the reactivation or valorization of waste as first or second raw materials. In such a context, the new behavior is required in the management of waste" [15].

For the second step, the data collection was carried out by the regional academic Spin-Off data collection. In particular, the Lazio Region identifies within the S3, 6 areas of specialization, recognizing a potential and an advantage pursued through a series of interventions. The objecting is to foster a repositioning of industrial areas, driving the region innovation along a path of internationalization. The areas are: Aerospace and security; life sciences; cultural heritage; digital creative industries; agri-food; Green Economy.

We select for the study Green Economy because is the area that replies to the needs of the Circular Economy, in a more explicit way. The context in which it works is characterized by the following sub-areas: "Secure, Clean and Efficient Energy (Energy efficiency; Smart grid); Climate Action, Environment, Resource Efficiency and Raw Materials (Safety and monitoring of the territory); Food Security, Sustainable Agriculture and Forestry, Marine and Maritime and Inland Water Research" (Safety and land monitoring) and "Smart, green and integrated transport" (Security and Mobility Control).

After these steps, the authors in the third part they conducted an iterated and comparative analysis. In this regard, we investigate the potential of the business model of the companies to activate the factors for the circularity. The outcome of our analysis is the presence or absence of the "Closed loop", "Open Loop". Closed loop is defined by the Standard as closed cycle where products, components or materials are reused or recycled into the same or similar products or components. Open loop refers a multiple cycle where products, components or materials are reused or recycled in other cycles. Finally, a judgement about the ability to activate "Circular Economy" approach is provided. At this regard, for the proposed evaluation, we follow the definition provided by Kirchherr, et al (2017): "A circular economy is a transformative economy redefining production and consumption patterns, inspired by ecosystems principles and restorative by design, which increases resilience, eliminates waste and creates shared value through an enhanced circulation of material and immaterial flows" [4].

The document adopts a qualitative research approach based on the descriptive protocol of the case study defined by Yin [15]. It is configured as an exploratory study based on the documentary analysis of Silverman [16,17]. For the multiple comparative analysis, the Spin-Off companies have been selected from the list official provided by Netval (2019), classified as active and with operating and/or registered office in the Lazio Region. This has made it possible to extract 96 companies located in the territory regional, without excluding any product categories. The subsequent analysis was aimed at identifying companies in relation to the type of link, direct or indirect, with the themes of environmental aspects from the point of view of the production of goods or the provision of services. In this 53 companies outside the identified scope could be discarded, 13 companies were found not to be available, 5 ceased and 1 company was in liquidation. In this way. It was possible to obtain a dataset of 24 companies for analysis.

We adopted official documents released by companies, their mother Universities, official documents. Furthermore, the research concentrated the analysis on the services offered, using web or social media tools. The data identifies the formal referenced properties. By means of a conceptualization process we compiled a concept table for emerging scenarios (see Appendix A), identifying the factors defined above. All the four-author analyzed separately the selected Spin-Offs, based on the proposed categorization. After a first phase, the authors processed the information iteratively, to arrive at

a discussion of the scenarios identified, excluding non-pertinent information, formulating a univocal judgment on the identified outcome.

These permitted to reconstruct the picture of the critical factors of the Circular Economy activated by each Spin-Off company. A representation of the research process was shown in Figure 1.

The literature has confirmed a gap concerning the role that Academic Spin-Off companies can have within the CE context. This lack is confirmed by a research of the flow of the scientific production inside CE and CBM. The research objective starts from the need to fill it, and understand if and how Spin Off companies can be a Driver for the development of a circular business.

The first two steps are functional to the definition of the boundaries of the work. The third contributes to the description of a comparative analysis of the critical factors used by the Spin-Off Companies, in an Open loop or closed loop system. It is an initial attempt to define the context for the application of the CE. In each phase the qualitative research methods, reported above, are used for data collection and analysis. In the fourth phase the scenarios within the Spin-Offs operate are defined, and their role as a Driver for the CE is discussed.

## 3. Theoretical Background

Analyzing the international scientific literature of recent years, it is immediately evident that the attention to the topic Circular Economy and Eco-Innovation tools has been increased. The scientific production on the themes "Circular Economy", "Spin-Off" and "Circular Business Model" are numerous.

The role of the Circular Business Model is widely discussed in the literature (see for example [18–22] identified as models that can help to promote the transition. There are several examples of the application of these models that highlight the benefits deriving from the transition and the definition of new business objectives, such as the extension of the life cycle of the product, the closure of the production cycles, the high valorization of by-products or the redesign of the product design or the increase of Production efficiency. The European Union has a long-standing commitment to issues of sustainability, environment, health and safety that he has repeatedly attempted to convey, directly and indirectly, through its development programs. The transition to new models, capable of to bring about a competitive change can be seen in the commitments or in the adoption of the Circular Economy policies by the European Commission (EC). An example of this commitment is the adoption of the CE policies, by the European Commission (EC), in the "Circular Economy Package" that included in 2014 "Towards a Circular Economy: A zero waste program for Europe" and in 2015 the Communication "Closing the loop and EU action plan for the Circular Economy" or from the EU Framework Program for Research and Innovation research Horizon 2020 (European Commission, 2015).

Overcoming the linear approach is a fundamental requirement to support the transformation towards a model of Circular Economy. This certainly represents an element of business growth, activating opportunities, promoted through the activation of new models that provide for the application of the factors and principles underlying this model [23–26]. Repeating the same analysis for the keywords "Circular Business Model" the results are very poor and mainly consist in "Circular design" and investigated the relationship between CE, Big data and Internet of Things (IOT) [27].

The aspects related to the transition incorporate concepts such as value proposition, supply chain, the customer interface, which presuppose changes in the financial models or in the management of sustainability as a development factor. The transversal and inter-sectoral logic are no longer sufficient to redefine the bases of a sustainable. A systematic management system is necessary to base the change on the use of new key success factors. The balancing or alignment of stakeholders in the pursuit of a growth, oriented towards the creation of value, requires a systemic vision on three levels, product/process design, business model, and consumer behavior, combined with other enabling factors, such as policies, training actions, technology, and tools financial. Everything aimed at fully exploiting the potential of this new approach. Change can be promoted through the adoption of new business models, which can be the real driver for change [25]. These models take on different

characteristics, depending on the type of links within the cascading cycles that help to define the appropriateness of the exploitation of resources in a given context. These cycles change as a function of the bonds, and having the characteristic of narrow or long cycles, and provide for the development of a cascading chain for integrate the concepts of resource economy and sustainability into an operational framework which promotes the efficiency and appropriateness of exploitation. Overcoming logic linear through the adoption of tight cycles helps to maintain and exploit the integrity of a product, compared to the complexity, use of energy or the production of environmental impacts [28,29]. In long cycles, the activation of consecutive cycles replaces the flow of virgin raw material, counteracting the dispersion of the material out of the cycle itself with benefits in terms of cost and price, and consequently environmental impacts. The immediate consequence of the transition concerns the opportunities for the application of the new Circular Business Models. Both in the case of the adoption of new design models that the combination of products and services is a narrow or long cycle, there are consequences and opportunities for the creation of added value and for innovation of products and services in a circular key. In literature, a plethora of different scenarios that propose business management focuses on the generation or maintenance of value for the company or for the flow of raw materials, for the customer and in general for the stakeholders concerned (see for e.g. [22,25,30]). The transition to new approaches, such as PSS [31–33], also help to change the management of the use of the asset, replaced by a mix of services [34,35].

In this context, the basic principles of the Circular Economy are based on the following assumptions the transition of these models [36], although the borders of these aspects have not yet been clearly defined, due to a lack of uniformity in the interpretation of the approach. Ellen MacArthur identifies in this the preservation and improvement of natural capital, the optimization of natural resources, the resource yields in use and promoting the effectiveness of the system. Additional authors integrate these aspects by enhancing the eco design, instrumental in the search for solutions from the design phase [37], or on how to this insists on guidelines that should be incorporated into the design for the product increases its circularity [38,39]. The industrial symbiosis is also a very useful tool for the exchange of waste of production/energy and water and for the creation of the necessary networks for development of the Circular Economy [40,41]. Involves the development of territorial systems geographically localized, such as eco-industrial parks, districts and production networks whose complex interactions of resources aimed at activating benefits. economic and environmental [42–45]. The 3R principle, *recycle, reduce and reuse,* is a central aspect of the application of the Circular Economy [46,47], favouring the emphasis on further principles based on the use of the suffix "R": Recovery (4R) [48], Reclamation (5R) [49], Repurpose (6R) [46], until recycle is considered materials (7R), Recovery energy (8R), re-mine (9R) [50].

The results of this review analysis was given an exhaustive overview of the scientific relevance of Circular economy, model and design and contribute to identify research gaps and to provide potential future research directions on the topic. Therefore, the study addresses the following research question: What role for Spin-Offs? They could contribute to the Circular practiced development as a driver?

Only 2 relevant scientific papers were identified:

1. *Lybæk, R., & Kjær, T. (2017). Enhancing identified circular economic benefits related to the deployment of the Solrød biogas plant. Engineering and Applied Science Research, 44(2), 97–105* [51].

2. *Stahel, W.R. (2007). Sustainable Development and Strategic Thinking. Chinese Journal of Population Resources and Environment, 5(4), 3–19* [52].

The concept of the Academic Spin-Off, in fact, has been for a long time the object of definitions heterogeneous not attributable to a single interpretation or a clear measure. The Commission has also taken the initiative to establish a legislative framework to clarify its meaning, but it has been the result of the consolidation of practices, more or less recognized or shared by universities. The establishment of Spin-Off companies is, without doubt, an alternative instrument to promote the transfer of knowledge and technologies to the commercial and productive sector. It is an opportunity for to

stimulate competition, promote the economic development of a territory, create employment, generate wealth and make the most of the results of university research. The way in which these companies are establishing themselves in the international context, thanks also to the new entrepreneurial and commercial role assumed by the Universities, makes these companies as an important reference point for innovation management and system competitiveness productive. In the literature, the theme of innovation management through Spin-Off companies has aroused significant international debate on the desirability of choosing the best organizational forms, capable of growing and conveying scientific research towards model's development complexes. In particular, the field of Spin-Off has been extensively studied and addressed not only to understand the added value compared to the corresponding start-up companies ventures [53], but especially with regard to the ability to generate positive performances, contributing directly or indirectly to the generation of profitability [54]. In general, these companies follow development models based on the interconnection of internal and external factors for the exercise of business activity [55]. Among the factors that in the literature have an effect on performance are of these companies we can cite the "Incubation and support", the "Knowledge and skills", "Financial Resources", "Social Relations and Networks", "Dimension", "Innovation", "Localization", "Responsibility and trust" and "Motivation" [12].

## 4. Results and Discussions

With the spread of the paradigm and the institutional impetus (at least European), the evolution of the studies and analysis tools on the Circular Economy are considerably increased. In particular, circular strategy, circular business models and circularity factors. Considering business model strategies, scholars mainly focus on studying closing material loops strategy, while slowing the loops, which requires a radical change of consumption and production patterns, is only marginally included with respect to CE implementation [27] are still not well defined and it is not clear the link between Spin-Off companies and their role in the context of the Circular Economy. On the application of Spin-Offs for the Circular Economy, the international literature only highlights two scientific works.

The first investigates how the experiences arising from the development of the biogas plant of Solrød in Denmark, a large scale centralized biogas plant, can help the future biogas technologies to achieve circular economic benefits [51]. Includes considerations on future developments through the scientific research of research bodies related to the park industrial, highlighting circular developments of "closed loops". The second, more wide-ranging, analyses from an economic and social point of view, the impact of technological development in overcoming resource scarcity through the recovery [52]. Recovery technologies are generally developed by research Spin-Offs. In this case could lead the circular paradigm back to an "open loop" vision.

All the research units considered for the analysis are characterized by a high level of heterogeneity compared to the areas of the S3 regional areas, which include cross-sectoral sectors: Food Security, Sustainable Agriculture and Forestry, Marine and Maritime and Inland Water Research and Smart, green and integrated transport and Climate Action, Environment, Resource Efficiency and Raw Materials and Secure, Clean and Efficient Energy.

The Academic origin and the location within the S3 of the Lazio Region contributes in reinforcing the expectations with respect to the ability to bring added value to the production context of the regional territory. The cases were selected to identify, study and discuss the factors that influence the activation of business models in a circular economy logic.

Also here, as in the literature we can distinguish two lines of intervention: a first line closed loop where the activities take place within a closed system, usually sectoral, and where the recovery and/or valorization of the waste takes place through internal reuse or recycle into the same or similar product. The material can be subjected to an intermediate treatment in order to allow for reuse as secondary raw material in the internal production system or in the external (open cycle). In this case a wide range of sectors foster the cyclic enhancement and the reuse or recycle into an alternative flow.

Analyzing through the exploratory analysis our sample catalogued in Table 1, it appears four characteristics about the Spin-Offs analyzed, depending on the nature of the product/services offered, and on the key factors identified. They can be traced back to four possible scenarios within an input-output Matrix (Figure 2), starting from the aggregation of the activities with respect to the output generated by the companies. The Matrix shows the results of the analysis. The factors have been investigated within the double configuration of our outputs, that consider the presence or absence of cycle loop (the closed loop (CL) and Open Loop (OL), and the ability to move toward Circular Economy. We get four possible scenarios, characterized by the intersection of outcome: Presence or Absence of Closed Loop or Open Loop, and from the Presence or Absence of Circular Economy. For each scenario we have identified the analyzed factors and the companies that distinguish it.

**Table 1.** Spin-Off Activities.

| Spin-Off Observed | Activities | Characteristics |
|---|---|---|
| a | Use and sharing of measurements and data for the energy and environment sectors | High customized services inspired to sustainability principles |
| b | Research and development of nanostructured electrode materials. Storage solutions with Lithium batteries. | |
| c | Support to the integrated management of environmental services for environmental reclamation and remediation environmental requalification | |
| d | Design and construction of photovoltaic systems; supply of components for photovoltaic systems; design of anchoring systems for photovoltaic systems | |
| e | Use of radar and optical data to create thematic maps and maps of electromagnetic and hydrocarbon pollution | |
| f | Design and engineering and IT services in the field of energy efficiency and sustainability | |
| g | Support in land management in the field of waste and pollution | |
| h | Provision of services and marketing of highly innovative products in primary agricultural, mechanical, food and public administration enterprises | |
| i | Agroforestry Environment al and territorial management and monitoring | |
| j | Development of software for energy and environmental impact assessment of sustainable transport systems | Reduction of environmental impacts |
| k | Most suitable technical solutions in the field of renewable energy | |
| l | Design and construction of energy plants from renewable sources | |
| m | Production of energy generation and microgeneration systems | |
| n | Monitoring of industrial renewable energy plants | |
| o | Extraction of energy from vibrations, electromagnetic fields and other forms of energy | |
| p | Fast urban and suburban road charging service for Electric Vehicles. | |
| q | Monitoring and treatment through software, automations and sensors of drinking water, control of air and $CO_2$ quality using NDIR technology in greenhouses | |
| r | Research and training in sustainable agriculture and social inclusion | Sustainable approach to reduce impact and promotes ethic values |
| s | Design and implementation of devices for internal, external and extreme environmental monitoring | |

**Table 1.** *Cont.*

| Spin-Off Observed | Activities | Characteristics |
|---|---|---|
| t | Research and development of solutions for the use of cyanobacteria and microalgae; promotes the sustainable use of algal biomass and environmental biotechnology and applications for cultural heritage | Reduce by high valorization of waste |
| u | Re-use and exploitation of waste materials from the pharmaceutical industry for the production of biomaterials for the creation of medical devices; provides technical and scientific assistance to the development of innovative processes in the field of environmental technologies; design of integrated electronic and photonic systems; | |
| v | Development of innovative processes for the recovery of metals from primary and secondary raw materials | |
| w | Toxicology, Pharmaceutical and Industrial Chemistry, Food Safety and the Environment. | |
| x | Design and implementation of radio frequency devices and sensors for environmental monitoring and within building materials | |

Source: Authors' Elaboration.

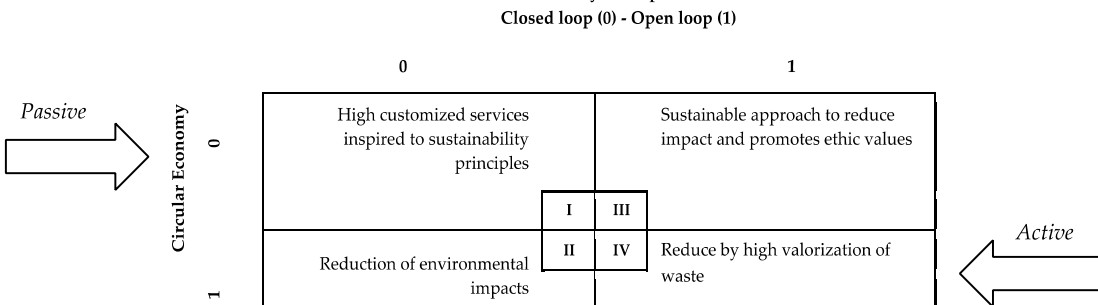

**Figure 2.** Input-Output Matrix. Source: Our Elaboration on Spin-Off analysis.

In the first scenario the emergent characteristic is the "High customized services inspired to sustainability principles". The services that are offered by the spin-off companies recall the university vocation, related to research and training. The focus is on innovative technological solutions, mainly linked to a service and not to a product. The vocation towards sustainable principles is strong (amplified by their industry). The reduction of environmental impacts is achieved by supporting partner companies in the management of their processes. The "reduce" factor is carried out using advanced monitoring systems that allow reductions in energy consumption and a reduction in environmental impacts. Overall, the role of spin-off companies does not allow to stimulate or activate a cascade cycle, within a closed loop or behaviors that show effects on the production context in a circular way.

The second scenario appears more complex, where the focus is the "Reduction of environmental impacts" using the factors "service support", "reduce" and "repurpose". The "services support" remains the central element of the Spin-off activity, mainly aimed at offering advanced solutions for the design or for the implementation of monitoring and control services, aimed at improving the environmental performance of companies. These activities have internal and external effects in customer companies, contributing to activate an additional factor, the "reduce" the impact of energy, using innovative solutions to manage and optimize energy sources. The "sharing" of data contribute the consolidation of the business and consequently the adoption of innovation among users. Only one unit (unit q) differs from the services offered. Here, product orientation to reduce environmental impact in agriculture, in a closed system (cycle loop). Overall, the activities promoted by Spin-Off companies contribute to improving the sustainability of the industry, projecting it towards a virtuous

approach to the application of the CE. The "Sustainable approach to reduce impact and promotes ethic values", included in the third scenario, works in a loop system. The attention to sustainable of their activity is considered and allows to bring out another factor, that is ethic values. In their activities promote environmental liability and social inclusion, but is unable to activate the CE.The "service support" continues to be the key aspect for the Spin-Offs. The "reduce" factor is connected to the monitoring of environmental impact, which support activity for pollution reduction as a result of the redistribution of energy.

The fourth is characterized by an open loop system that works in a circular economy perspective. As for the other scenarios, the "service support" is the base line of the activities. It offers customized or advanced solution to intervene on the environmental impact. At the same time, it allows the introduction of another factor, not considered in the model, represented by the eco-design for environmental efficiency. As a result, the characteristic that emerges is the "high valorization of waste", which starts the business to an Open loop system, activating additional factors, such as "reduce" factor for the environmental impact, and "reuse" factor, for the recovery of resources. In this scenario the Spin-Offs become key players in promoting or activating circular processes.

An in depth analysis shows us, that there are two particular contexts, "Passive" and "Active". In the first one, although the enterprises have skills and sustainable technologies, they are not able to activate cascade systems to switch toward Circular Economy (I scenario). Furthermore, companies suffer the context in which they work, for example it is the case of the Spin-Off that supply high customized services (e.g., III Scenario) and not scalable. They don't affect the market with their innovation, and consequently don't give the input to the switch to CE.

The second refers the active context. It is present within the II e IV scenario, thanks to the ability to activate CE. Eco-design emerges as an instrumental factor in the search for sustainable solutions, from the design phase [37,56], to the assembly or the disassembly phase [39], or to support the "design or architecture" for the efficiency and the sustainability of the business [57].

However, here the discriminating factor is the "High valorization of waste". This characterizes the spin-off companies as driver for Circular Economy. It provides a propulsive thrust, activating new opportunities for the use or the development of new complementary activities. This will allow to define a new Cascade process, which will contribute towards switching to CE.

## 5. Conclusions

In most of the international projects and case studies, there are practices of industrial symbiosis related to open cycles and a strong cooperation between actors, related to the exchange of energy and matter.

From the analysis of the literature, however, there is a gap since there are no specific indicators, created specifically and verified; and there is no massive incentive intervention that pushes to specific interventions in the cases analyzed.

Therefore, the need to fill this gap, starting with the integration of the present study with a further deepening and extension of the sample, should be realized in order to identify the key variables for the realization of a general framework to be followed.

The result shows that the Academic Spin-Offs can be a driver of Circular Economy, as long as that fall within the IV scenario, described above, is characterized by the interconnection of an open loop system that works in a circular approach. The "High valorization of waste" represents the discriminant in this scenario, which allows to activate a cascade system in a multi-stakeholder perspective [15].

The implications that emerge from our work, for the researcher and policy maker, concern the incentives for the creation of Spin-Off companies, within the third mission of the universities.

Many Universities have started a business development through the economic enhancement of their innovations by private owned companies with the ability to move easily within the market. These are activities that directly interact with society, providing a scientific contribution and accompanying traditional teaching and research missions.

In literature the importance of the two traditional missions of the universities is highlighted, but at the same time it opens up new issues in the management of these dimensions, in light of the changes that have taken place at the international level [58]. In this renewed vision, the creation of a Spin-Off company undoubtedly represents an alternative tool to promote the technology or knowledge transfer to the commercial and production sector. Likewise, it is possible to use these tools as drivers for the application of CE. For this purpose, an implication concerns the possibility of using incentive policies capable of addressing/using the third mission as development drivers for the application of the Circular Economy.

The scenarios proposed suggest implications for the continuation of the study of spin-offs as drivers for the circular economy. In this regard, future research concerns the study of how the incentive policies of universities affect the start-up of Spin-Off companies to make them a driver of sustainability and circularity.

Additionally, the work suggests the extension of the analysis to all the sectors identified in the regional Smart specialization strategy, although much remains to be done from the point of view of the study of such behaviors over time.

Among the main limitations of this analysis, in fact, there is a limitation in the sample of Spin Off analyzed and a lack of clear references in the literature such as to identify a set of specific indicators on which to base a generalized model.

Finally, the configuration of these indicators and the consequent characteristic factors within a business model would make it possible to define in a structured manner a development strategy that can help to describe the company's performance within the market, but fails to describe the contribution offered in terms of Circular Economy.

On these aspects too, however, the reference literature is still lacking, preventing the formulation of hypotheses about the role that these societies are assuming in the transition to the new model. This study, in its preliminary phase, tries to overcome this gap to investigate how these innovative companies can contribute to the pursuit of the transition to the new model.

The strength of this analysis is that, however, the information collected at the moment is not systematized in any general model in which a procedure is formalized in which the development of indicators and the demand for research is at the moment central to the more general discussion on the development of the paradigm of the Circular Economy.

**Author Contributions:** Conceptualization, S.P. and G.A.; methodology, S.P., G.A.; validation, E.M.M.; formal analysis, S.P., G.A. and E.M.M.; resources, S.P.; writing—original draft preparation, S.P., G.A.; writing—review and editing, E.M.M., M.A.d.T. All authors have read and agreed to the published version of the manuscript.

**Funding:** This research received no external funding.

**Conflicts of Interest:** The authors declare no conflict of interest.

# Appendix A

**Table A1.** Scenario 1: Spin-Off in a closed loop system and absence of the CE approach.

| Spin Off | Service Support | High Waste Valorization | Share | Reduce | Reuse/Redistributed/Repurpose |
|---|---|---|---|---|---|
| a | Customized solutions based on specific customer needs | | | Innovative measurement and control solutions for energy and the environment | |
| b | Improvement of product solutions | | Use and sharing of technologies developed and patented in the energy and automotive sectors for lithium-ion batteries | Innovative measurement and control solutions for energy and the environment | |
| c | Environmental monitoring, safety management, landscape analysis and management services | | | Support for pollution reduction | |
| d | Support as to the management of energy resources and plant design and maintenance | | | | Redistribution of electricity from renewable sources produced by third parties |
| e | Support for the creation of thematic maps using radar and optical data with maps for pollution monitoring | | | | |
| f | Innovative solutions for energy saving (software and hardware), monitoring of energy consumption dynamics | | | | |
| g | Support in land management in the field of waste and pollution | | Construction of innovative application services on Earth Observation data | Monitoring and reduction of environmental impacts (definition of indicators) | |
| h | environmental management solutions aimed at defining indicators and reducing impacts | | | Energy monitoring, data analysis and strategic planning. | |
| i | Management and monitoring of the environment and agricultural resources, infrastructure, watercourses, quarries, landfills; precision agriculture in the wine sector, monitoring and forecasting of production and crop quality, monitoring of plant diseases in agriculture and forestry; forest fires. | | | | |

\* The "Life Extension" factor and "Refurbish/Remanufacture/Recondition" factor are not present in any company considered; therefore they have been eliminated from the table.
Source: Authors' Elaboration.

**Table A2.** Scenario 2: Spin-Off in a closed loop system and presence of CE approach.

| Spin Off | Service Support | High Waste Valorization | Share | Reduce | Reuse/Redistributed/Repurpose |
|---|---|---|---|---|---|
| j | Software solutions for sustainable energy and transport, modelling and simulation of energy systems and their impact on the environment and population | | | Reduction of the impacts of energy systems, energy saving and efficiency | |
| k | Consultancy and design in the field of innovative engineering; third generation organic solar cell photovoltaics | | | Reduction of the impacts of energy systems, energy saving and efficiency | |
| l | Design of photovoltaic, wind and bio-mass systems Certification of the manufacturability of wind power systems | | | Reduction of the impacts of energy systems, energy saving and efficiency | |
| m | Design of energy plants technological solutions for industry, transport and biomedical | | | Power generation and micro-generation systems | |
| n | Consulting and monitoring services in the field of industrial energy, development of cloud data sharing software | | Cloud data sharing | Energy monitoring, data analysis and strategic planning. | |
| o | Development of technological solutions for energy harvesting with energy extraction from vibrations or electromagentative fields | | | Monitoring of the electrical infrastructure; | |
| p | Sustainable mobility consultancy for fast charging of urban and suburban electric motors | | Recharging systems for electric columns | Reduction in the use of non-renewable energy. Sustainable mobility | |
| q | Environmental control services creation of sensors for the control and monitoring of C02 and air quality | | | Integrated nanotechnology for sustainable sensing water and sanitation. Sustainable use of agricultural waste and by-products. Reduction of environmental impact | |

* The "Life Extension" factor and "Refurbish/Remanufacture/Recondition" factor are not present in any company considered; therefore they have been eliminated from the table.
Source: Authors' Elaboration.

**Table A3.** Scenario 3: Spin-Off in an open loop system and absence of the CE approach.

| Spin Off | Service Support | High Waste Valorization | Share | Reduce | Reuse/Redistributed/Repurpose |
|---|---|---|---|---|---|
| r | Advanced solutions for agricultural sustainability, environmental responsibility and social inclusion | | | | |
| s | Energy and sensor design applied to the monitoring of indoor, outdoor and extreme environments | | | Monitoring of environmental parameters in standard measuring ranges and a series dedicated to the measurement of parameters in extreme environments | |

* The "Life Extension" factor and "Refurbish/Remanufacture/Recondition" factor are not present in any company considered; therefore they have been eliminated from the table.
Source: Authors' Elaboration.

**Table A4.** Scenario 4: Spin-Off in an open loop system and presence of the CE approach.

| Spin Off | Service Support | High Waste Valorization | Share | Reduce | Reuse/Redistributed/Repurpose |
|---|---|---|---|---|---|
| t | Customized solutions based on specific customer needs | Valorization of algal biomass | | Reduction of energy costs towards the production of algal biomass. Sustainable solution for wastewater treatment plants | |
| u | | Valorization of special waste in biomaterial | | Innovative measurement and control solutions for energy and the environment | Reuse of waste materials from the pharmaceutical industry for the production of biomaterials and the creation of innovative medical devices |
| v | Technological solutions for advanced waste recovery | Treatment exhausted accumulators and batteries, or metals from electrodes | | | Recovery of waste and reuse as second feedstock (from WEEE) |
| w | Support to the reduction of environmental impacts in the agri-food and industrial sectors. | use of lignin for innovative applications (sun screen, bio plastics, Nano particles) | | Reduction of environmental impacts through pollutant degradation technologies | Recovery of natural bioactive substances present in wastewater from agro-industrial processes (helio-technical, coffee and hazelnut production), reuse of lignin |
| x | Design of radio frequency devices and sensors for environmental monitoring and within building materials | | | Use of RFID technologies, to collect data and monitor food and drug environments; and also in logistics, transport and shipping. | |

* The "Life Extension" factor and "Refurbish/Remanufacture/Recondition" factor are not present in any company considered; therefore they have been eliminated from the table.
Source: Authors' Elaboration.

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
