# Peer review of "Entrepreneurial Drivers for the Development of the Circular Business Model: The Role of Academic Spin-Off"

_sustainability, doi:10.3390/su12010423_

Round 1

Reviewer 1 Report

The article presents the first results from a larger research project on the role of academic spin-offs for the circular economy. It includes a literature review and a qualitative data collection on relevant academic spin offs. In the following I present my comments to the article:

The paper stats at several places that it presents the first results from a larger research project. However, this information in not relevant because the paper should be of high value in itself and not only have value in connection to your project. My main comment connects to the comment above. When starting a new research project you of course have to do a literature review to get to know the topic. However, that literature review is mainly for your own knowledge development. It is very seldom the case that such literature review has value to be published in an academic journal. You even state several times that there is no literature on the topic yet (you only find two papers) and in your paper you do not explain the need for a literature review. The gaps you present only motivate exploratory case studies. In the paper (in appendix 1) you include information about case companies, but there is no valuable analysis done of that data. The main body of your paper is based on a literature review of circular economy and circular business models and the two articles on the topic. Having your purpose in mind you need to do a exploratory case study based on Yin (which you state) but that automatically includes a review of relevant literature which is presented in relation to your purpose not as a literature review itself. But the main part of the paper should be based on the data analysis for the cases and results should come from there. As of know you conduct either the literature review or the case study properly. Your title talks about entrepreneurial drivers. However, that term does not appear in the rest of the paper and no entrepreneurial drivers are identified. Your objective stats that you will identify factors make spin-off companies able to realize the CE. But even these factors are very hard to grasp from your paper. You mention some factors, but it is not clear where there come from and what role they play. You say that you did a survey. The term survey is very closely associated with quantitative studies. Please be more specific why you talk about survey and in which way the study was quantitative or qualitative. You have a large part on circular business models. I know that CBM are important, but it is not clear how CBMs are relevant for this paper. How would results be different without considering CBMs? The background of literature need to be connected to your purpose. This would mean that spin-offs need to be part of that in some way. There are issues in the outline of the paper that does show that you did not carefully revised the paper in the end. E.g. section 3.1. has no heading and the first sentence of section 5 should not either be part of the manuscript.

Author Response

Authors’ response to Review 1

We first want to thank the Reviewer for the valuable comments and careful handling of the manuscript. The comments gave us different useful insights and were very helpful for clarifying the contributions of the study and improving the quality of the paper. In the revised version of the manuscript, we have addressed each issue raised by the Reviewer. This letter explains the changes we have made (Reviewers’ comment in bold, answers from the authors in plain text).

The article presents the first results from a larger research project on the role of academic spin-offs for the circular economy. It includes a literature review and a qualitative data collection on relevant academic spin offs. In the following I present my comments to the article:

Dear reviewer, thank you very much for your comments. Following, our response of your comments:

The paper stats at several places that it presents the first results from a larger research project. However, this information in not relevant because the paper should be of high value in itself and not only have value in connection to your project. My main comment connects to the comment above.

Thanks for the comment, this work is the result of extensive research work that is carrying out an interdisciplinary team of researchers

In fact, we have specified in the text the contextualization of the research by changing the introduction and better explaining the background.

When starting a new research project you of course have to do a literature review to get to know the topic. However, that literature review is mainly for your own knowledge development. It is very seldom the case that such literature review has value to be published in an academic journal. You even state several times that there is no literature on the topic yet (you only find two papers) and in your paper you do not explain the need for a literature review.

We defined the role of literature, used to better define the research question.

In the text we also specified the research framework through a figure and how the literature contributed to the formulation of the background and the detection of the gap

The gaps you present only motivate exploratory case studies. In the paper (in appendix 1) you include information about case companies, but there is no valuable analysis done of that data.

In the appendix we insert the analysis of the case studies, we observe how the spin-off companies behave with respect to the identified factors. We modify the table to make it more readable.

The main body of your paper is based on a literature review of circular economy and circular business models and the two articles on the topic. Having your purpose in mind you need to do a exploratory case study based on Yin (which you state) but that automatically includes a review of relevant literature which is presented in relation to your purpose not as a literature review itself. But the main part of the paper should be based on the data analysis for the cases and results should come from there.

Yes, we specify this aspect and change the literature review paragraph and the case studies analisys in the results section.

As of know you conduct either the literature review or the case study properly. Your title talks about entrepreneurial drivers. However, that term does not appear in the rest of the paper and no entrepreneurial drivers are identified.

We insert the entrepreneurial drivers and the spin off ecosystem description in the text

Your objective stats that you will identify factors make spin-off companies able to realize the CE. But even these factors are very hard to grasp from your paper. You mention some factors, but it is not clear where there come from and what role they play.

More attention has been paid to the explanation about the factors identified.

You say that you did a survey. The term survey is very closely associated with quantitative studies. Please be more specific why you talk about survey and in which way the study was quantitative or qualitative.

It is a qualitative analysis, the authors specify that in the text

You have a large part on circular business models. I know that CBM are important, but it is not clear how CBMs are relevant for this paper. How would results be different without considering CBMs?

We explain the role of CBM and Circular strategy in the context

The background of literature need to be connected to your purpose. This would mean that spin-offs need to be part of that in some way.

The background of literature need to be connected to your purpose. This would mean that spin-offs need to be part of that in some way.

Now the background is connect with the rest of the paper

There are issues in the outline of the paper that does show that you did not carefully revised the paper in the end. E.g. section 3.1. has no heading and the first sentence of section 5 should not either be part of the manuscript.

We have corrected typos and errors

Reviewer 2 Report

The authors recognize that the study is part of a larger study. I recommend that the study be completed before assessing its publication.

The results are inconclusive, given the shortage of the sample. The 

The article is not written properly, as it contains comments from the authors themselves, such as the introduction of the conclusion (line 291) about which they comment "This section is not mandatory but can be added to the manuscript if the discussion is unusually
long or complex ".

The literature review section does not specify how many articles have been reviewed, but only how many have been found in the databases.

Author Response

Entrepreneurial drivers for the development of the circular business model: the role of Academic Spin-Off

Manuscript-ID: sustainability-637775

Authors’ response to Review 2

We first want to thank the Reviewer for the valuable comments and careful handling of the manuscript. The comments gave us different useful insights and were very helpful for clarifying the contributions of the study and improving the quality of the paper. In the revised version of the manuscript, we have addressed each issue raised by the Reviewer. This letter explains the changes we have made (Reviewers’ comment in bold, answers from the authors in plain text).

The authors recognize that the study is part of a larger study. I recommend that the study be completed before assessing its publication.

Thanks for the comment, this work is the result of extensive research work that is carrying out an interdisciplinary team of researchers

In fact, we have specified in the text the contextualization of the research by changing the introduction and better explaining the background

The results are inconclusive, given the shortage of the sample.

We implement the results and explain better them.

The article is not written properly, as it contains comments from the authors themselves, such as the introduction of the conclusion (line 291) about which they comment "This section is not mandatory but can be added to the manuscript if the discussion is unusually

long or complex ".

It was a typo error. We have corrected typos and errors.

The literature review section does not specify how many articles have been reviewed, but only how many have been found in the databases.

We defined the role of literature, used to better define the research question.

In the text we also specified the research framework through a figure and how the literature contributed to the formulation of the background and the detection of the gap.

Reviewer 3 Report

(1) Abstract is too extensive; needs to be more streamlined, focused and compact.

(2) Methodology: Little justification of methodological approach, methods and design.

(3)Literature review: Lacks adequate logical links and lacking in clarity. Limited application to business and management practice. 

(4) In pages 5 and 6 long paragraphs do not facilitate reading.

(5) Paper contribution or implications not clear.

(5) Appendix A, Table 2 is not readable; its usefulness is limited.

Author Response

Entrepreneurial drivers for the development of the circular business model: the role of Academic Spin-Off

Manuscript-ID: sustainability-637775

Authors’ response to Review 3

We first want to thank the Reviewer for the valuable comments and careful handling of the manuscript. The comments gave us different useful insights and were very helpful for clarifying the contributions of the study and improving the quality of the paper. In the revised version of the manuscript, we have addressed each issue raised by the Reviewer. This letter explains the changes we have made (Reviewers’ comment in bold, answers from the authors in plain text).

Abstract is too extensive; needs to be more streamlined, focused and compact.

It was streamlined

Methodology: Little justification of methodological approach, methods and design.

We explain better the methodology and introduce a Framework design

Literature review: Lacks adequate logical links and lacking in clarity. Limited application to business and management practice. 

Thanks for your suggestion. We specify this aspect and change the literature review paragraph and the case studies analisys in the results section

In pages 5 and 6 long paragraphs do not facilitate reading.

We changed this part.

Paper contribution or implications not clear.

More attention has been paid to these aspects

Appendix A, Table 2 is not readable; its usefulness is limited.

We change the table and now they are more readable.

Round 2

Reviewer 1 Report

Your paper has significantly improved, but a better response on how you adressed the review comments would have been appriciated.

It is very positive that you moved away from the literature review description and focus on the case studies. However, to be able to really understand your analysis and the conclusions you draw from it, the activities described in table 1 would need another analysis step to form aggregared dimensions. Now it is very detailed activities that should be group in more generic categories to understand the connection to table 2 (which is a figure) or your scenarios need to be labled inorder to give an understanding of what this could be. However, to include CE yes/no as one of the parameters seems not very relevant as all companies in some way contribute to a more circular economy even though the company itself not is circular. This makes your analysis very weak.

Author Response

Entrepreneurial drivers for the development of the circular business model: the role of Academic Spin-Off

Manuscript-ID: sustainability-637775

Authors’ response to Review 1

We first want to thank the Reviewer for the valuable comments and careful handling of the manuscript. The comments gave us different useful insights and were very helpful for clarifying the contributions of the study and improving the quality of the paper. In the revised version of the manuscript, we have addressed each issue raised by the Reviewer. This letter explains the changes we have made (Reviewers’ comment in bold, answers from the authors in plain text).

It is very positive that you moved away from the literature review description and focus on the case studies. However, to be able to really understand your analysis and the conclusions you draw from it, the activities described in table 1 would need another analysis step to form aggregared dimensions.

Now it is very detailed activities that should be group in more generic categories to understand the connection to table 2 (which is a figure) or your scenarios need to be labled inorder to give an understanding of what this could be.

Thanks for the comment. We have aggregated the characteristics in the table 1, as suggested, and modifying the figure (ex table 2) considering the changes made.

However, to include CE yes/no as one of the parameters seems not very relevant as all companies in some way contribute to a more circular economy even though the company itself not is circular. This makes your analysis very weak.

Thanks for the comment. More attention has been paid to the explanation about the role of spin-off about CE.  Furthermore, we changed the concept table in Appendix to make it easily readable

Reviewer 3 Report

Overall progress has been made from previous submission.

Appendix A should be more effectively designed as it is not easily readable.

Author Response

Authors’ response to Review 3

We first want to thank the Reviewer for the valuable comments and careful handling of the manuscript. The comments gave us different useful insights and were very helpful for clarifying the contributions of the study and improving the quality of the paper. In the revised version of the manuscript, we have addressed each issue raised by the Reviewer. This letter explains the changes we have made (Reviewers’ comment in bold, answers from the authors in plain text).

Overall progress has been made from previous submission.

Appendix A should be more effectively designed as it is not easily readable.

We changed the concept table in Appendix to make it easily readable

Round 3

Reviewer 1 Report

Well done revision!